# The Ying and Yang of Sphingosine-1-Phosphate Signalling within the Bone

**DOI:** 10.3390/ijms24086935

**Published:** 2023-04-08

**Authors:** Kathryn Frost, Amy J. Naylor, Helen M. McGettrick

**Affiliations:** Rheumatology Research Group, Institute of Inflammation and Ageing, University of Birmingham, Birmingham B15 2TT, UK

**Keywords:** osteoblast, osteoclast, S1P, bone mineral density, bone, osteoporosis

## Abstract

Bone remodelling is a highly active and dynamic process that involves the tight regulation of osteoblasts, osteoclasts, and their progenitors to allow for a balance of bone resorption and formation to be maintained. Ageing and inflammation are risk factors for the dysregulation of bone remodelling. Once the balance between bone formation and resorption is lost, bone mass becomes compromised, resulting in disorders such as osteoporosis and Paget’s disease. Key molecules in the sphingosine-1-phosphate signalling pathway have been identified for their role in regulating bone remodelling, in addition to its more recognised role in inflammatory responses. This review discusses the accumulating evidence for the different, and, in certain circumstances, opposing, roles of S1P in bone homeostasis and disease, including osteoporosis, Paget’s disease, and inflammatory bone loss. Specifically, we describe the current, often conflicting, evidence surrounding S1P function in osteoblasts, osteoclasts, and their precursors in health and disease, concluding that S1P may be an effective biomarker of bone disease and also an attractive therapeutic target for disease.

## 1. Introduction

Despite its static appearance, bone is a highly dynamic tissue that is continually adapting to mechanical strain and damage. This bone remodelling process crucially involves two cell types (osteoblasts and osteoclasts) that communicate with each other within temporary bone modelling units (BMU) [1] and respond to changes in the local microenvironment to maintain homeostatic levels of bone formation and resorption. Disruption of this delicate balance can occur with age and in a variety of inflammatory disorders and cancer, leading to abnormal bone mineral density and increased risk of fracture [2]. Sphingosine-1-phosphate (S1P) is a member of the sphingolipid family that has previously been described to effect multiple cell functions such as survival [3], proliferation [4], and differentiation [5]. There is a growing body of evidence that indicates its importance in the bone pathology of several inflammatory diseases [6], and, more recently, its role in the maintenance of bone homeostasis throughout the life course has been identified. Here, we detail the current understanding of the different, and sometimes opposing, roles of S1P in bone remodelling and how these can appear divergent depending on the local milieu. We discuss the evidence that S1P impacts the onset and progression of bone diseases and identify areas where further research is required to ensure that the therapeutic potential of S1P can be effectively harnessed.

## 2. Bone Remodelling

Bone remodelling has two crucial functions in adults: the repair of damaged bone and the strengthening of existing bone tissue. In response to activating stimuli, such as mechanical strain, osteoclast precursors migrate to the bone surface where they undergo osteoclastogenesis [7]. Osteoclasts are large multinucleated cells that originate from the fusion of monocyte/macrophage haematopoietic lineage cells [8]. Mature osteoclasts directly mediate bone resorption through their release of chloride and hydrogen ions, which dissolve the alkaline hydroxyapatite mineral component of the bone [9]. The remaining collagen matrix is broken down via osteoclast-secreted lysosomal enzymes, matrix metalloproteinases (MMPs), and cathepsin K [10,11]. Following resorption, mesenchymal stem cell (MSC)-derived osteoblasts [12] replace the resorbed tissue by secreting osteoid, a type I collagen-rich organic extracellular matrix [12]. Over a period of several weeks, this matrix is mineralised with hydroxyapatite crystals to complete the bone remodelling process [13].

Bone remodelling is tightly regulated by the crosstalk between osteoblasts and osteoclasts and local environmental cues (damage, mechanical strain and hormones, e.g., parathyroid hormone—PTH), which all impact the balance of mineralisation and resorption to maintain homeostasis [14,15,16]. Many of the therapies for bone conditions stem from our understanding of the factors influencing osteoblast and osteoclast precursor migration, osteoblastogenesis, or osteoclastogenesis, which have enabled the enhancement of bone formation (e.g., PTH therapy) or the prevention of bone destruction (e.g., bisphosphonates). Despite these successes, there is an unmet clinical need for therapies that can prevent bone loss and repair damaged bone tissue. One newly identified potential therapeutic target is S1P.

## 3. Sphingosine-1-Phosphate

S1P is part of the sphingolipid metabolic pathway that also includes sphingosine and ceramide (Figure 1). Each metabolite has been shown to be involved in the survival, migration, and proliferation of osteoblast and osteoclast progenitor cells (reviewed by [17]). S1P is metabolised from ceramide via two reactions. First, ceramidase (CDase) converts ceramide into sphingosine in a reversible reaction with ceramide synthase (CerS) (reviewed by [18]). Sphingosine kinases (SPHK1 and SPHK2) then convert sphingosine to S1P. The phosphorylation of sphingosine can be reversed in the presence of sphingosine phosphate phosphatases (SPPase), or S1P lyase can irreversibly break down S1P into ethanolamine phosphate and hexadecenal [19]. Collectively, these reactions regulate extra- and intra-cellular sphingolipid concentrations, and thus the functional consequence of their signalling on the cell and surrounding cells (extensively reviewed by [20]).

## 4. S1P Signalling

S1P has both intracellular and extracellular functions in a variety of tissues (reviewed by [21]). One accepted principle within bone and other tissues is the sphingosine rheostat: here the intracellular levels of ceramide and S1P influence cell survival and proliferation, with ceramide being considered pro-apoptotic and S1P acting in an anti-apoptotic fashion [22]. No studies have looked at this rheostat directly in osteogenic cells; however, various studies have shown the similar actions of both ceramides and S1P. Additionally, there are few studies that directly investigate the role of intracellular S1P within bone cells [23,24]; therefore, reviews on the subject tend to infer functions based on other studies performed in other cell types.

### 4.1. Intracellular S1P Signalling

Current work investigating intracellular S1P within bone focuses on the action of the enzymes required for its production (SPHK1 or 2) rather than the bioactivity of S1P itself. Alterations in SPHK1 activity and/or expression result from altered autocrine signalling of the extracellular S1P receptors (S1PR1-5) due to the secretion of S1P, rather than altered intracellular S1P signalling [25]. Indeed, alterations in SPHK1 expression during osteoblast differentiation have been linked to the activation of S1PR1 signalling [23]. To the best of our knowledge, no studies have investigated the impact of intracellular osteoblastic SPHK1 and/or S1P signalling; however, it has been shown that the osteoclast-specific deletion of SPHK1 has no effect on bone mass in male or female mice [26]. Similarly, siRNA knockdown of SPHK1 in a murine macrophage cell line (RAW264) or in primary murine bone-marrow-derived macrophages had no effect on osteoclast maturation in response to RANKL [24]. Conversely, the upregulation of SPHK1, through the retroviral transfection of HA-tagged SPHK1, resulted in a moderate decrease in osteoclast maturation [24], suggesting that SPHK1 in osteoclasts acts as a negative regulator of osteoclastogenesis. The authors suggested that RANKL-mediated osteoclast activation results in an upregulation of SPHK1, which regulates the osteoclast function through the inhibition of c-FOS and NFATc1 via p38-dependent signalling [24]. Crucially, these differences were not observed in the presence of extracellular S1P, suggesting that they are mediated by either the intracellular action of SPHK1 or intracellular S1P [24]. These findings demonstrate the importance of intracellular S1P on osteogenesis and bone resorption and underscore the need for additional investigations into the intracellular receptors of S1P and their function in osteoclasts.

The inhibition of SPHK2 in the murine macrophage cell line RAW264 resulted in a reduction in c-FOS expression and subsequent osteoclastogenesis [27], identifying intracellular SPHK2 signalling as essential during the initial stages of osteoclastogenesis. In contrast, SPHK2^−/−^ osteoclasts had normal resorptive activity in vitro, despite the observation that global SPHK2^−/−^ mice have reduced trabecular bone mass [26]. This finding led to the in vivo phenotype being attributed to alterations in osteoblast formation, which was demonstrated by a reduction in collagen 1 gene expression and a significant impairment of the anabolic response to PTH within these mice [26]. Crucially, the impact of SPHK2 on PTH signalling was not determined, making it difficult to draw any firm conclusions about the exact role of intracellular S1P signalling in this process. These studies both postulate a role for SPHK2 in altering osteoclast or osteoblast function to drive changes within the bone architecture. One thing to consider with these studies is the fact that they focus solely on the intracellular kinase activity to infer the intracellular S1P function. However, increases in kinase activity leads to increased S1P production and therefore secretion, meaning differences could be due to autocrine signalling rather than intracellular activity. It is possible to overcome this issue by inhibiting SPSN2, thereby blocking S1P secretion; however, the interpretation of these results will be confounded by the assumptions of the sphingosine rheostat. Further research into the intracellular actions of S1P is urgently needed to enable the identification of potential intracellular targets that inhibit intracellular S1P function.

### 4.2. Extracellular S1P Signalling

In comparison to intracellular signalling, significantly more research has been conducted on the role of extracellular S1P signalling in the bone. Given the limited knowledge surrounding the S1P intracellular targets in bone, this review focusses predominantly on the functional consequence of the autocrine and paracrine actions of extracellular S1P. These include its actions directly on the cells within the bone and the mechanisms by which altered S1P cell–cell signalling results in impaired bone remodelling.

S1P is secreted by osteoblasts and osteoclasts, predominantly through the spinster 2 (SPNS2) transporter [28,29], allowing binding to one of the G-coupled protein S1PR. Osteoblasts, osteoclasts, and their precursors express four types of S1PR (S1PR1, -2, -3, and -4), which vary in expression levels across the maturation/differentiation status of the cells [17] and are thought to play critical roles in bone cell migration, differentiation, and communication (Figure 2).

## 5. S1P in Bone Precursors and Resident Cells

### 5.1. S1P as a Chemoattractant and Chemorepellent for Osteoclast Precursors

Osteoblast precursors (mesenchymal stem cells, MSCs) and osteoclast precursors (monocytes) are recruited into the bone tissue during periods of remodelling (reviewed by [30]) in response to CXCL12-CXCL4 and S1P-S1PR signalling (reviewed by [31]). S1P is detected in many different tissues; however, under homeostatic conditions, blood holds the highest concentration of S1P due to the secretion from red blood cells, thrombocytes, endothelial cells, and platelets (reviewed by [32]). Within blood, S1P is found bound to albumin and lipoproteins, particularly HDL. Studies have demonstrated that S1P bound to albumin or HDL does not impact its ability to activate S1PRs, but rather creates a reservoir of S1P, keeping the S1P within circulation (reviewed by [32]). As total S1P is detected at higher concentrations in blood [33] and at lower concentrations within the bone [34], a concentration gradient is created along which osteoblast and osteoclast precursors migrate. The importance of this gradient in precursor migration was demonstrated when S1P breakdown was inhibited by either the inhibition or deletion of S1P lyase in vivo, causing the gradient to dissipate, resulting in increased bone mass and strength coupled with decreased osteoclast number [35]. The next sections will discuss the mechanisms and signalling pathways in which these actions are carried out.

### 5.2. Osteoclast Progenitors

Osteoclast precursor chemotaxis/migration has been shown to be dependent on the local S1P concentration: at low concentrations (<10^−7^ M), osteoblast precursors (murine macrophages, RAW624.7) efficiently move through a Transwell filter towards S1P [36]. However, at higher concentrations (10^−6^ M), S1P repels the RAW624.7 cells, resulting in greater numbers remaining in the filter unable to move [36]. This observation raises a key question: how does S1P simultaneously attract and repel osteoclast precursors (monocytes)?

Osteoclast precursors express two of the S1P family of receptors, S1PR1 and S1PR2 [36]. siRNA knockdown of S1PR1 reduced osteoclast precursor migration at low S1P (10^−9^ M) concentrations, a phenomenon not observed following S1PR2 knockdown, indicating that S1P chemoattraction is mediated by S1PR1 [36]. Similar observations were made in wild-type mice treated intravenously with the S1PR1 agonist, SEW2871 (which causes receptor internalisation and mimics the inhibition of S1P activity), where increased numbers of osteoclast precursors were detected in the bloodstream [37], suggesting S1PR1 is responsible for keeping cells within circulation, where the S1P concentration is greater. Conversely, S1PR2 knockdown in osteoclast precursors (RAW264 cell line) enhanced migration towards S1P in vitro irrespective of S1P concentration, suggesting that S1PR2 negatively regulates osteoclast precursor migration [37]. Similarly, the administration of the S1PR2 antagonist, JTE013, increased monocytes in the circulation in vivo. Thus, S1PR2 is responsible for retaining osteoclast precursors within the tissue [36] or promoting migration from the circulation to the tissue. Collectively, the studies from this group have demonstrated opposing functions of S1PR1 and S1PR2 expressed by osteoclast precursors through the facilitation of chemotactic responses to S1P, thus regulating the migration of osteoclast precursors.

As previously described, RANKL is required for the differentiation of osteoclast precursors and is secreted from both osteoblasts and osteocytes (reviewed by [38]). Additionally, RANKL may also influence osteoclast precursor migration via its actions on S1PR1 [39,40]. For example, the intraperitoneal injection of RANKL induced a severe osteoporotic phenotype in mice that was reversed in the presence of the S1PR2 antagonist, JTE013 [39,40], indicating that RANKL directly impact S1P-related chemotaxis of osteoclast precursors. Therefore, RANKL appears to have a dual role in enhancing bone remodelling through (i) promoting the differentiation of osteoclast precursors and (ii) facilitating their migration into the bone.

### 5.3. Osteoblast Progenitors

S1P has also been investigated for its effect on osteoblast precursors (MSC). Primary murine MSCs express gene transcripts for all five of the S1PRs [41], whereas human bone-marrow-derived MSCs were only shown to express S1PR1-3 transcripts [42,43]. Of note, an additional amplification revealed an extremely low-level expression of S1PR4 and S1PR5 transcripts in human MSC [42], indicating that they are expressed at much lower levels compared to the murine MSC. It should be noted that the expression of S1PRs have also been shown to be influenced by culture density (3000–5000 cells/cm^2^ vs. 15,000–20,000 cells/cm^2^), where a marked increase in S1PR1 and S1PR3, but not S1PR2, gene expression was detected in the high-density cultures [42]. Therefore, alterations in cell culture protocols and crosstalk between cells in culture may explain the conflicting findings within the literature. Unfortunately, the exact seeding density or plate sizes used are not always reported in sufficient detail within the current literature to establish this.

Similar to that described for the osteoblast progenitors, MSCs also display a concentration-dependent response to S1P [43] where human BM-MSC migrate towards low concentrations (1–10 nM) of S1P in a Transwell migration assay, whilst no migration was seen at higher concentrations (50–1000 nM) [43]. These data suggest that an S1P gradient is necessary for MSC migration. The opposing migratory properties of S1P on MSC movement have been, in part, attributed to the downstream signalling mediated by the different S1PRs; for instance, at low S1P concentrations, S1PR1 and S1PR3 downstream signalling occurred via Gi/ERK and supported migration, which was sensitive to S1PR1/3 or ERK1/2 pharmacological inhibitors in Transwell experiments [43]. By contrast, S1PR2 signalling via the G_12/13_-RhoA-ROCK pathway inhibited migration, whilst the pharmacological inhibition of ROCK with Y27632 allowed for an increased migration at 1 nM [43]. The JTE-013-mediated inhibition of S1PR2 or genetic deletion (S1PR2^−/−^) also inhibited migration in MSC scratch wound and Boyden chamber assays [41].

Despite the evidence above that the activation of S1PR2 results in the chemorepulsion of MSC in response to S1P, conflicting data indicate a role for S1PR2 in enhancing MSC migration via signalling crosstalk with S1PR1 [44]. Quint et al. showed that the pro-migratory response of the FAK/PI3K/AKT signalling is lost when JAK/STAT activity is inhibited, suggesting that the S1PR2-mediated activation of the FAK/PI3K/AKT signalling pathway induces the activation of S1PR1-mediated JAK/STAT signalling [44]. These data suggest that S1PR2 works in concert with, rather than against, S1PR1 to increase MSC migration.

The differences in S1P receptor roles in migration may be due to the origin of the progenitors (e.g., blood or tissue resident) and the experimental concentration of S1P used to activate the receptor. Unlike osteoclast progenitors, which often originate in the blood, the majority of MSC that differentiate into osteoblasts for bone remodelling are derived from the bone marrow or periosteum. These cells are not subjected to the high concentrations of S1P found within the blood. As such, the strength of any concentration gradient along which MSC are migrating is substantially reduced compared to that experienced by osteoclast progenitors. Moreover, the Quint study demonstrated that osteoclast-conditioned media acts as a chemoattractant to MSCs, increasing chemotaxis in an S1P-dependent manner (where migration was blocked in the presence of S1PR antagonists), thus suggesting that osteoclasts can recruit MSCs to the site of resorption [44]. Given that MSC responses to S1P vary dependent on S1P concentration, researchers should take care to use biologically relevant concentrations for their tissue of interest. One hypothesis is that smaller gradients in S1P concentrations are responsible for MSC chemotaxis to the bone surface opposed to those responsible for MSC recruitment from the blood.

## 6. Role of S1P in Mature Bone-Resident Cells

### 6.1. Osteoclasts

Osteoclasts are multinucleated cells involved in bone matrix demineralisation by releasing protons via vacuolar H^+^-adenosine triphosphatase (H^+^-ATPase), resulting in acidification of the resorption compartment and the release of calcium and phosphates (reviewed by [45]). The differentiation of monocytic osteoclast precursors into osteoclasts occurs within the bone remodelling compartment, where several molecules (M-CSF, RANKL, and OPG) drive osteoclastogenesis (reviewed by [46]). The most well-studied of these factors, RANKL, appears to at least partially elicit its osteogenic response via S1P signalling.

Following RANKL-induced osteoclastogenesis, there is an enhanced gene expression and the activity of SPHK1 and SPHK2, resulting in increasing in both intracellular and extracellular S1P [24]. Intracellular S1P was assessed by comparing extracellular S1P levels in culture media with concentrations within the cells (cell lysates) [24]. Mature osteoclasts express tartrate-resistant acid phosphatase (TRAP); therefore, the expression of this enzyme is used as a marker of differentiation. SPHK1-induced intracellular S1P inhibited TRAP expression, as did the overexpression of SPHK1, whilst the siRNA knockdown of SPHK1 increased TRAP [24]. Ryu et al. demonstrated that RANKL increases the expression and also activation of p38 and ERK MAPKs in osteoclast precursors to promote osteoclastogenesis [24]. In cells treated with SPHK1 siRNA, there was an increase in p38 protein expression compared to those treated with control siRNA, whereas in SPHK1-overexpressing Mx-HA-SPHK1 cells, there was a significant decrease in p38 [24]. From these data, it is postulated that the SPHK1 upregulation and activation seen in response to RANKL results in increased intracellular S1P to induce a negative feedback loop, suppressing p38 signalling in osteoclast precursors and allowing the tight regulation of osteoclast differentiation [24] (Figure 3). Collectively, these studies suggest RANKL-mediated osteoclastogenesis is regulated though intracellular S1P; therefore, alterations in osteoclast S1P responses may result in osteoclastogenesis being unregulated, causing increased bone resorption. An enhanced understanding of how intracellular S1P inhibits p38 may enable the identification of new therapeutic targets to inhibit osteoclast formation and reducing bone formation.

The literature examining the effects of extracellular S1P on osteoclast precursors is both limited and conflicting: one study showed that exogenous S1P has no effect on osteoclastogenesis in bone marrow macrophages (BMM) in culture [24], whilst we have observed the increased differentiation of a murine macrophage-like cell line, RAW264.7, in response to S1P (unpublished observations). One possible explanation for this conflict could be the fact that the expression pattern of S1PR1 and S1PR2 alters during BMMs differentiation [24,48] but not during RAW264.7 differentiation [24]. That being said, the concerted actions of RANKL and S1P appear to be important for BMMs osteoclastogenesis [40]. The S1PR2 antagonist (JTE013) inhibited the formation of TRAP-positive osteoclasts and the absence of activity/bone resorption pits [40]. Therefore, S1PR2 appears to be essential for both osteoclastogenesis and osteoclast function [40]. Hsu et al. further demonstrated that S1PR2 regulates osteoclastogenesis via the regulation of podosome-adhesive proteins that are required for the fusion of monocytes to form multinucleated osteoclasts [40]. siRNA knockdown of S1PR2 reduced the levels of the protein kinases, p-PI3K, p-SRC, or p-PYK2, required for the adhesion and fusion of monocytes [40]. These data suggest that extracellular S1P and its receptors may play a role in osteoclastogenesis; however, further research is required to understand the differences in responses to exogenous S1P between studies and how other S1PRs may impact osteoclast formation.

### 6.2. Osteoblasts

The expression of S1PRs appears to vary during osteoblast differentiation; however, which receptors are expressed at what stage of differentiation remains somewhat contradictory in the literature. It is frequently reported that S1PR1-3 are the most highly expressed [49]; more contentious is whether S1PR4 is also expressed [29]. Our analysis of publicly available mouse bone marrow stromal cell RNASeq datasets (GSE128423) allowed the interrogation of osteoblast populations at distinct stages of maturation. Early osteoblast progenitors, pre-osteoblasts, and mature osteoblasts were detected by looking at osteoblast differentiation genes, such as COL1A1, RUNX2, and BGLAP1. Only S1PR1, S1PR2, and S1PR3 were identified to be expressed in immature osteoblast populations, and their expression decreased as osteoblasts mature, with no detection of S1PR4 at any stage (unpublished observations). S1PR1 and S1PR3 expression appeared to increase during the initial stages of osteoblastogenesis [17], but may wain following differentiation (unpublished observations). Differences in S1PR expression on osteoblasts during maturation may account for differences in the responses of osteoblasts to S1P.

S1P has been reported to act as an osteoanabolic, increasing osteoblast differentiation and proliferation [41,50,51,52,53,54]. Hashimoto et al. demonstrated that the addition of exogenous S1P to early osteoblast progenitors (C3H10T1/2 pluripotent stem cells) resulted in the increased expression of genes linked to osteoblast differentiation, e.g., LRP5/6, ALPL, and BGLAP [50]. Additionally, S1P increased mineralisation (Alizarin red staining), suggesting it promotes MSC-like cells to differentiation towards an osteoblast lineage [50,51]. A recent study into the effect of S1P on murine dental pulp stem cells, which display a MSC-like phenotype and express S1PR1 and 2, showed that low levels of osteoblastic differentiation can be induced in response to S1P at various concentrations (1–10 µmol-L^−1^) [47]. These cells also displayed decreased expression of several osteoblast-specific genes (including ALP, RUNX2, and COL1A1), as well as decreased mineralisation following 3 weeks of culture with S1P [47]. The differences between these two studies may stem from differences between the two cell types used, with the S1PR expression on C3H10T1/2 yet to be reported.

Studies have also looked at S1P in pre-osteoblasts and osteoblast-like cell lines [47,52,53]. Murine MC3T3-E1 cells are an immature osteoblast cell line that matures in response to S1P [47,53]. Matsuzaki et al. first showed that S1P increased the protein expression and activation of PI3K/Akt/GSK-3β signalling at increasing concentrations of S1P (0.1–2 µM), resulting in increased ALP activity (p-nitrophenyl phosphate substrate levels) and increased Alizarin red-stained mineral [53]. Increased mineralisation in response to increasing concentrations of S1P (1–10 µmol-L^−1^) was replicated in MC3T3-E1 cells by Choi et al. [47]. SaOS-2 cells, a human osteoblastic osteosarcoma cell line, also show increased PI3K/Akt/GSK-3β signalling in response to S1P stimulation, which resulted in enhanced ALP levels and mineralisation [53]. The addition of S1P also increased the proliferation of primary rat calvarial osteoblasts, as measured using [H^3^]-thymidine incorporation [52]. Unfortunately, this study did not look at markers of differentiation, so no comments on this can be made. In our hands, S1P reduced ALP activity and Alizarin red staining, thus the differentiation of primary mouse calvarial osteoblasts (unpublished observations). By contrast, we were unable to detect the increases in osteoblast maturation and mineralisation previously reported using MC3T3s. These disparities may be due to differences in the osteogenic media used as both Higashi et al. and Choi et al. supplemented with steroids (dexamethasone or hydrocortisone), whilst neither Matsuzaki et al. nor we did [47,53].

Of all the S1P receptors expressed by MSCs and osteoblasts, the most well-studied in the context of osteoblastogenesis is S1PR2. Isolated murine BMSC from wildtype mice treated with JTE-013 or S1PR2^−/−^ mice exhibited an enhance proliferation as assessed using an MTA assay [41]. Moreover, these cells had lower levels of the osteoblast differentiation marker osteopontin compared to wildtype untreated cells [41], suggesting S1P/S1PR2 signalling enhances osteoblastogenesis and that the inhibition of S1PR2 retains osteoblasts in a prolonged proliferative state.

Members of the SMAD signalling family are required for osteoblastogenesis [53], where SMAD4 forms heteromeric complexes with phosphorylated SMAD1/5/8, resulting in nuclear translocation and regulation of gene expression (e.g., RUNX2 [55]). At high S1P concentrations (2 μM), downstream signalling through ROCK and SMAD1/5/8 signalling led to enhanced RUNX2 gene expression and increased von Kossa staining (a measure of mineralisation) in the MC3T3 murine osteoblastic cell line [50]. These data suggest that S1PR2 is required for the differentiation and maturation of osteoblasts. By contrast, others have shown that S1PR2 inhibition increased adipocyte genetic markers FABP4 [41], suggesting that whilst S1PR2 promotes MSC differentiation, its function is not necessarily osteogenic-lineage specific.

Knockout mice have been used to investigate the function of S1PR1/3; however, S1PR1 global knockout in mice is embryonic lethal [50]. Osteoblast-specific S1PR1 deletion results in viable pups that have no detectable differences in bone formation at either 3 or 8 months [29]. Conversely, S1PR3 global knockout mice are viable and exhibit no developmental defects at 3 months; however, osteopenia is seen at 8 months [29]. This S1PR3^−/−^-induced osteopenia is a result of a reduced bone formation rate demonstrated by reduced BV/TV (a measure of bone volume) and BFR/BS (a measure of bone formation over time), whilst parameters linked to resorption, e.g., serum osteocalcin(OcN)/bone morphogenic protein (BMP) and collagen degradation products (carboxy-terminal collagen crosslinks: CTX) remained unaffected [29]. Furthermore, when S1PR3^−/−^ calvarial and bone-marrow-derived osteoblasts were cultured in the presence of S1P, there was a reduction in mineralisation compared to wildtype cells, suggesting reduced osteoblast maturation, although no other measurements of osteoblast maturation were assessed [29].

Although S1PR1 appears to have no role in post-natal bone remodelling, S1PR3 signalling is required for continued remodelling following skeletal maturation in mice, i.e., from mid-life onwards. Brizuela et al. demonstrated that SPHK1 mRNA expression increased with osteoblast maturation and that the inhibition of SPHK1 using an inhibitor (SK-II) or by the sequestration of S1P with a neutralising antibody reduced osteoblast maturation (as identified through reduced alkaline phosphatase activity and RUNX2 mRNA expression) [28]. A similar reduction in maturation was observed when the S1PR1/3 antagonist (VPC23019) was applied, but not when using specific S1PR1 (W146) or S1PR2 (JTE013) antagonists [28]. Taken together, these data suggest that S1P is essential for osteoblast maturation via S1PR3. As such, S1PR3 agonists may offer a therapeutic benefit in the treatment of catabolic bone disease in adults, but they are not expected to offer the same benefits to children.

## 7. Role in Osteogenic Communication

To regulate bone remodelling, osteoclasts and osteoblasts secrete a variety of factors that attract, activate, and inhibit each other (reviewed by [56]). This crosstalk is essential for normal bone remodelling, and if communication is disrupted, it results in altered bone formation and resorption. There is some evidence of the impact of S1P on this bidirectional crosstalk. Culturing naïve or activated macrophages with murine bone-marrow-derived stem cells (BMSCs) was sufficient to induce the upregulation of SPHK1 and S1PR1 mRNA expression [23], suggesting that macrophages secrete molecules that induce SPHK1 and S1PR1 expression in the bone marrow niche. Additionally, co-culturing-activated macrophages with BMSCs resulted in increased RANKL production that was dependent on the activation of S1PR1, where the secretion of RANKL was lost following the siRNA knockdown of S1PR1 [23]. The authors postulate that osteoblast–osteoclast signalling activates S1PR1 and SPHK1, resulting in autocrine S1P/S1PR1 signalling that triggers increased RANKL production, which subsequently stimulates osteoclastogenesis within the co-culture [23] (Figure 3). This example highlights the complexity of S1P communication between osteogenic cells and emphasises the importance of such co-culture experiments.

The molecular mechanisms by which S1P acts as a communication molecule within bone is complex due to its varying roles in migration and differentiation and the sensitivity of its receptors to a fluctuating S1P concentration. The S1P-induced differentiation of MSCs into osteoblasts triggers the release of RANKL and subsequently the osteoclastic release of S1P [24]. This feed-forward mechanism allows the coupling of osteoblasts and osteoclasts, which is important for the regulation of bone remodelling. S1P has also been shown to stimulate RANKL expression in osteoblasts through the upregulation of COX2 via ERK and p38 MAPK pathways [24]. However, a model in which S1P purely stimulates the osteoblast release of RANKL, and its subsequent stimulation of osteoclastogenesis, does not explain the increased bone mass seen in response to S1P lyase inhibition or deletion [35]. The activation of osteoblast-expressed S1PR2 has been reported to cause the release of the soluble RANKL decoy molecules, OPG, via p38-GSK3β-β-catenin and noncanonical WNT5A-LRP5 signalling [23,35,54]. Given that S1PR2 has a lower affinity for S1P compared to S1PR1 [57], one hypothesis is that at low concentrations of S1P S1PR1/3, activation induces RANKL release. RANKL serves to activate osteoclasts and promote a further S1P release. As S1P concentrations increase, there is a shift in S1PR activation towards S1PR2, resulting in an alteration in the RANKL/OPG ratio in favour of OPG, which reduces bone resorption rates. An alternative explanation is that increased S1P results in S1PR1 internalisation, thereby reducing RANKL production and altering the RANKL/OPG ratio in favour of bone formation (Figure 3). Collectively, S1P bioavailability regulates the activation and the cessation of bone resorption.

Calcitonin has been shown to impact osteoclast–osteoblast communication via S1P. Calcitonin downregulates the gene expression of the S1P transporter, SPNS2, on primary murine osteoclasts [29] and is a negative regulator of bone formation. Calcitonin receptor knockout mice (CALCR^−/−^) have increased trabecular bone volume and evidence of increased bone formation rate (serum alkaline phosphatase levels are increased) [29]. No effect on bone resorption markers (e.g., collagen breakdown products—CTX assay) were detected in these mice [29]. In response to calcitonin, Keller et al. observed that SPNS2 gene expression was downregulated in wild type osteoclasts [29]. This reduction in SPNS2 gene expression was coupled with a reduction in extracellular S1P, a difference that was not observed in the osteoclasts from CALCR^−/−^ mice. These data indicate that calcitonin reduces S1P secretion from osteoclasts via the inhibition of SPNS2. The phenotype was lost in CALCR^−/−^ mice when S1PR3 was knocked out in osteoblasts [29], suggesting that S1P secreted by osteoclasts in response to calcitonin acts to increase osteoblast maturation and mineralisation via S1PR3. Calcitonin acts to negatively regulate this circuit via the inhibition of osteoclastic secretion of S1P. Together, these data imply that calcitonin naturally regulates bone remodelling through altering S1P-mediated osteoclast–osteoblast communication.

As detailed above, the mechanisms by which S1P acts as a key regulator of bone remodelling (through communication between osteoblasts and osteoclasts) are starting to be uncovered. Challenges in the isolation and culture of osteocytes have limited research into how this S1P-mediated communication affects osteocytes. A recently developed osteocyte cell line (Ocy454 [58]) has been shown to increase the production of S1P in response to mechanical stimulation, which triggers the upregulation of SPHK1 and downregulation of SPPases (SGP11 and SGPP11) [59,60]. If confirmed in vivo, these data could indicate that S1P functions in the activation of the bone remodelling cycle, attracting osteoclast and osteoblast precursors to the site of injury and promoting osteogenesis within the bone modelling unit. These findings suggest the existence of a positive feedback loop, whereby S1P is produced by osteocytes in response to mechanical stimulation which acts directly on osteocyte S1PRs to further increase S1P release [60]. Moreover, intracellular levels of S1P enhance calcium and prostaglandin E2 signalling in osteocytes (Ocy454 [61]), raising the possibility that S1P influences other osteocyte functions. Exactly how extracellular S1P and S1PR expression on osteocytes impacts communication with other bone resident cells following activation is unclear and further research is urgently required.

## 8. S1P in Catabolic Bone Diseases

### 8.1. Osteoporosis

Osteoporosis is a common bone disorder affecting 23.1% of women and 11.7% of men aged 18–95 globally [62]. The disorder is characterised by a decreased bone mineral density and bone strength, increased risk of fracture, reduced quality of life, and increased morbidity [62]. Osteoporosis develops due to a disturbance in the balance between osteoclast-mediated bone resorption and osteoblast-mediated bone formation, resulting in excessive resorption and net bone loss [62]. Although both sexes can be affected by osteoporosis, with the incidence rising with age, postmenopausal women are at the highest risk (reviewed by [63]).

Kim et al. (2016) and Bae et al. (2016) observed that osteoporosis patients display a higher ratio of S1P in their plasma compared to bone marrow, with an additional study by Ardawi et al. (2018) finding a positive correlation between osteoporosis-related fracture and S1P plasma levels [64,65,66]. Increased plasma levels of S1P steepen the concentration gradient between the tissues, resulting in the increased migration of osteoclast precursors to the bone parenchyma. Osteoporosis patients show an increased expression of bone resorption markers, despite no alterations in bone formation markers [64,67,68]. This increase in resorption can be attributed to the enhanced migration of osteoclast precursors into the tissue [64,67,68] (Figure 4). A large population-based cohort by Weske et al. (2018) demonstrated a strong positive correlation between plasma S1P and bone formation markers (procollagen type 1 N-terminal propeptide (PINP) [35]. In contrast to the aforementioned studies, increased calcium and a reduced bone resorption (C-terminal telopeptide of type 1 collagen—CTX) were also correlated with high plasma S1P levels [68]. The paucity of data in all these studies on the levels of S1P in the bone marrow and parenchyma (and therefore the ratio between bone and blood) makes their interpretation challenging. Ahn et al. reported decreased levels of S1P within the bone marrow of patients who experienced an osteoporotic hip fracture, suggesting that the S1P gradient is steepened by both changes in plasma and tissue S1P [69]. However, due to the high rate of turnover of S1P in bone, the accuracy of these measurements is uncertain.

The reliability of the S1P measurement is also a limiting factor. S1P can bind to carrier proteins, such as HDL, LDL, and albumin. Tests vary in their ability to detect the bound lipid, leading to the inaccurate detection of S1P bioavailability. Song et al. reported that bound S1P has distinct functions based on the related carrier protein [65], where for that which is albumin-bound but not HDL-bound, S1P was associated with an increased risk of osteoporotic fracture, indicating that HDL acts as a scavenger protein to reduce S1P function. However, the effects of albumin-bound S1P alone were lost following corrections for confounding factors [65], with only total S1P showing a significant difference in fracture risk. Clearly, further investigation is required to understand the involvement of S1P carrier proteins and the deleterious effects of S1P in osteoporosis.

As osteoporosis is most prevalent in postmenopausal women, substantial research has been performed investigating the association between S1P and oestrogen. Oestrogen plays an essential role in the development and maintenance of bone, having a variety of indirect and direct effects on formation and resorption [63]. The production of S1P has been linked to the activity of 17β-oestradiol (E2), and both S1P and oestrogen have been shown to enhance osteoblast proliferation [70]. E2 increased SphK1 mRNA expression and protein expression in human osteoblasts, whilst S1P caused an upregulation in ERβ but not Erα mRNA, as well as an increase in SPHK1 and S1PR1 [70]. It is likely that fluctuations in oestrogen may result in alterations within S1P signalling, but further research is required to identify whether the use of S1P signalling modulators could act as osteoporosis therapeutics.

The risk of osteoporosis is very strongly correlated with ageing, yet few studies have looked specifically at the effect of age on S1P as most participants were postmenopausal women [64,65,67,68,71,72]. The studies that have used patients undergoing hip replacement surgery [67,69] often have smaller sample sizes and predominantly include female participants who are over 50. Moreover, they also frequently fail to report sex differences. Although sex is recorded, it is often adjusted for rather than comparative data being presented; therefore, any sex differences in the role of S1P in bone are lost within the data. A recent review discussed the involvement of S1P in the dysregulation of nutrient sensing, cellular senescence, and proteostasis with age, and all these factors could impact bone remodelling [73]. Although there is substantial evidence of the role of S1P in age-related disease, such as osteoporosis, the role of age-related changes in S1P expression on bone function is required in order to establish whether it has any bearing on age-related decline in bone density.

Taken together, these data suggest that S1P could act as a potential biomarker for osteoporosis, allowing an earlier intervention prior to severe bone loss. Moreover, S1P poses an interesting therapeutic target for osteoporosis through the use of S1PR2 inhibitors, which would reduce bone resorption by partially inhibiting osteoclast precursor migration. Despite these exciting clinical benefits, considerable further research is required as highlighted here, especially dissecting the causal link between changes in S1P signalling within the bone with age and/or sex and increased risk of osteoporosis in postmenopausal women and the elderly.

### 8.2. Paget’s Disease

Paget’s disease of bone (PDB) is a chronic and progressive focal skeletal disorder [74] which can manifest as a monostotic (singular bone) or polyostotic (multiple bone) phenotype [75]. Typically, this disease affects adults over the age of 55, resulting in increased bone fragility and bone pain in the affected area [76]. PDB skeletal lesions are caused by increased bone resorption coupled with disorganised bone formation [77] and altered vascular infiltration. Histologically, PDB patients have increased numbers of osteoclasts, which are larger and display higher levels of activity compared to osteoclasts from healthy controls [78].

Although it is the altered morphology and activity of osteoclasts that drives pathogenesis in PDB, the increased secretion of molecules, such as S1P, from these osteoclasts also results in an altered osteoblast function [79]. It has been recently shown that osteoclasts from both PDB patients and the Measles virus nucleocapsid protein (MVNP) mouse model of PDB display an increased expression of S1PK1, resulting in increased S1P production [79]. Additionally, osteoblasts from PDB patients and MVNP mice express increased S1PR3, with no differences in the expression of S1PR1 and S1PR2, when compared to osteoblasts isolated from healthy controls [79]. An in vitro co-culture of osteoclasts and osteoblasts from MVNP mice showed an increased S1PR3 expression compared to control mice [79]. The activation of the S1PR3 receptor with the S1PR3 agonist VPC24191 enhanced the osteoblast differentiation and expression of RUNX2 and COL1A1 [79]. The S1PR3 antagonist VPC23019, which decreased levels of differentiation genes, such as Osterix, was observed [79]. These data suggest the involvement of S1P in Paget’s disease, with a particular emphasis on the role of S1PR3, indicating a potential target for a future therapeutic to inhibit the dysregulation of osteoblasts. Further studies are required to identify which molecule(s) are mediating the upregulation of S1PR3 in osteoblasts. S1P is known to upregulate S1PR3 in other cell types, making it an obvious candidate for further investigation. Additional evidence revealing the link between S1P signalling within osteoblasts and osteoclasts and dysregulated function has the potential to identify novel therapeutic targets that might have clinical utility in PD in the future.

### 8.3. Inflammatory Diseases with Bone Damage

S1P has been known as a key driver of inflammation and is able to influence the migration of inflammatory cells and alter cytokine production (reviewed by [80]). Bone loss occurs both systemically and locally in several inflammatory diseases, such as rheumatoid arthritis (RA) and periodontitis (reviewed by [81]). In recent years, S1P has been investigated for its role in joint and bone inflammatory pathologies and mooted as a therapeutic target in several diseases.

#### 8.3.1. Rheumatoid Arthritis

Rheumatoid arthritis (RA) is a common autoimmune disorder characterised by persistent inflammation of the synovium and the destruction of cartilage and bone. One study, looking at the presence of S1P within the synovium of RA patients, showed a significant increase in S1P [82]. Lai et al. assessed the involvement of S1P in the collagen-induced arthritis (CIA) model of RA. The inhibition of SPHK1 by pharmacological (N-N-Dimethylsphingosine—DMS) or siRNA knockdown highlighted its involvement in joint destruction and inflammation [82]. DMS-mediated inhibition significantly reduced joint erosion and inflammation, reducing the incidence and severity of arthritis [82]. Similar observations were made using siRNA knockdown, which collectively suggested that an increased production of S1P via SPHK1 is required for the progression of inflammatory arthritis in the CIA model [82].

A large variety of cell types are involved in the pathogenesis of RA, and much of the research in this area has focused on S1P involvement in non-osteogenic cell types (recently reviewed by [83]). To the best of our knowledge, no one has investigated the action of S1P specifically on osteoblasts during RA pathogenesis. Current research has worked on the assumption that increased S1P within the synovium will increase the migration of osteoclasts to the bone surface to cause increased bone resorption [84].

There are a few studies that have reported alterations in bone parameters or bone cells in animal models of inflammatory arthritis [85,86,87]. One study did observe reduced bone erosion when FTY720, a high-affinity agonist of S1PRs, was administered to the SKG mouse (described by [88]) that spontaneously develops arthritis [86]. However, no other bone parameters were reported; therefore, it cannot be determined whether the reduced erosion was due to reduced osteoclast function or reduced joint swelling and inflammation. Only two papers have looked in detail at osteoclasts within an inflammatory arthritis model [85,87]. Using the FAS-deficient MRL/lpr model of spontaneous inflammatory arthritis, the authors highlighted significant bone loss due to an increased number of osteoclasts [87]. The analysis of mandibular condyle tissue showed an upregulation of SPHK1 and S1PR1 gene expression and an increase in the number of SPHK1- and S1PR1-positive cells in these mice compared to uninflamed control animals [87]. The addition of the NF-κB inhibitor SN50 reduced the number of TRAP-positive cells and attenuated the subchondral trabecular bone loss seen in MRL/lpr mice [87]. Additionally, SN50 decreased the number of SPHK1- and S1PR1-positive cells, suggesting that FAS and S1P/S1PR1 signalling act in concert with NF-κB to exacerbate pathogenesis in this model [87]. When hTNFα transgenic mice were crossed with SPHK1^−/−^ mice, a reduction in bone erosion was observed, with fewer mature osteoclasts in a location away from the bone margin being observed [88]. In the presence of excessive TNFα, SPHK1 or S1P production may be important for osteoclast maturation and activity, possibly due to changes in the S1P gradient that limit osteoclast precursor migration [85].

We also identified one clinical trial targeting the S1P pathway in RA (clinical trial: NCT00847886). The study involved LX3305, an inhibitor of S1P lyase that prevents the degradation of S1P. The inhibition of S1P lyase has been shown to increase bone mass [35] through the removal of the S1P gradient to reduce osteoclast precursors migration. The LX3305 phase 2 trial was completed in 2010; however, to date, the results of the trial have not been publicly reported.

#### 8.3.2. Periodontitis

Periodontitis is an inflammatory disease of the oral cavity which results in the destruction of the alveolar bone and other structures that support the teeth. Left untreated, periodontitis can cause severe pain and tooth loss. Analysis of a large population-based cohort showed that the serum concentration of S1P was greatly upregulated in patients with moderate and severe periodontitis compared to healthy controls [89]. Additionally, the SPHK1 protein expression was upregulated in gingival tissue samples from participants with periodontitis compared to healthy controls, suggesting that there are alterations in S1P metabolism in the context of periodontitis [89].

There are a number of identified risk factors for both periodontitis and RA: increasing age, female sex, smoking [90,91], and poor oral hygiene [92]. The effect of these environmental and biological risk factors on S1P production and function is likely to impact disease initiation and progression. Smoking alone has been implicated in the dysfunction of bone remodelling, resulting in reduced bone mass and bone mineral density, increasing the risk of osteoporosis [93]. The mechanisms surrounding this have been discussed at length in (review Al-Bashaireh et al., 2018 [93]). Overwhelming evidence suggests smoking can have direct effects on the bone by reducing osteogenesis and angiogenesis, as well as having indirect effects through alterations in antioxidant and cortisol production [93].

Research in other diseases (e.g., chronic obstructive pulmonary disease) has shown that smoking alters the S1P function by increasing levels of S1PRs and kinase activity [94,95]. Although the effect of smoking on S1P and other sphingolipids in the context of bone formation has not been formally studied, smoking has been shown to affect fracture healing [96,97]. Thymoquinone (TMQ) has previously been reported to increasing sphingolipid signalling in monocyte by increasing receptor mRNA expression [98]. Of relevance here, one study has investigated the effect of cigarette smoke on the efficacy of TMQ [97] in mice subjected to a model of bone healing. In this model, TMQ alone enhanced fracture healing, demonstrated by increased new bone formation, osteoblast number, and OPG:RANKL ratio [97]. Mice that received cigarette smoke (CS) alone or in addition to TMQ showed an increase in osteoblast number and OPG, with reduced RANKL expression [97]. Unfortunately, the mechanism of action of TMQ in relationship to the bone was not discussed in this study. Additional work is required to determine whether this is due to TMQ actions of S1PR expression, as previously shown [98].

Additionally, periodontitis is a risk factor for the development of RA [99], amongst other conditions, and, as such, aberrant S1P signalling within affected individuals could increase the frequency of comorbidity between these disorders and contribute to systemic bone damage. The above data suggest that S1P signalling is important within inflammation-mediated bone erosion. To understand the impact of S1P on bone formation and resorption in inflammatory disease, it is necessary to uncouple direct effects on the bone from the well-known effect of S1P on inflammation. Interventions that target S1P signalling will affect the bone and immune system, and separating the direct and indirect effects is extremely challenging. The current data suggest that there are alterations in S1P signalling in RA and periodontitis that drive inflammatory bone loss within these diseases. However, further research is required to unpack which are direct effects to enable the development of targeted therapeutics from the bone.

## 9. Conclusions

Increasing interest in the role of S1P in bone remodelling has revealed that it contributes to progenitor migration, homeostasis, and osteoblast/cyte–osteoclast communication. This review has shown that, although the field is in its infancy, there is a growing body of evidence indicating that S1P signalling is critical for bone homeostasis and that the disruption of these pathways can result in the dysregulation of bone remodelling. The notable actions of S1P in osteoblasts and osteoclasts are summarised in Table 1, along with their potential utility as therapeutic targets in the treatment of catabolic bone disease.

Gaps in the published data have led to the extrapolation of scientific dogma, such as the sphingosine rheostat, from other cell types, and this requires an urgent verification and examination of how the sphingosine rheostat within bone tissue alters S1P metabolism and how it may impact cell function and crosstalk. Going forward, the methodologies of any future study on this topic will need to be very carefully defined to allow an adequate interpretation of the data obtained. However, despite these limitations, the findings to date offer some hope that the S1P signalling pathway may be an attractive biomarker and/or therapeutic target for metabolic and inflammatory bone diseases in the future.

## Figures and Tables

**Figure 1 ijms-24-06935-f001:**
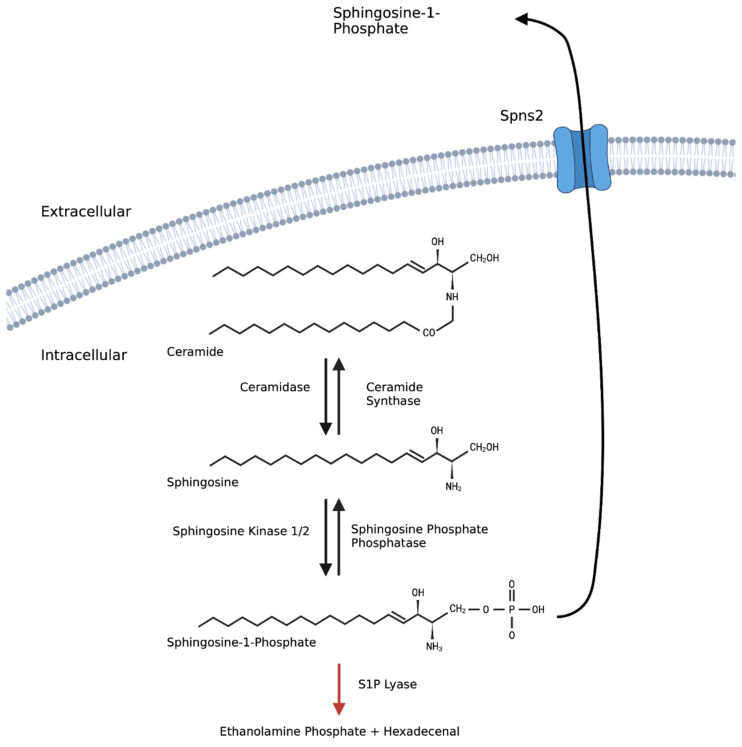
**Sphingosine metabolic pathway.** Ceramide, sphingosine, and sphingosine-1-phosphate can be metabolised into each other via various enzymes. Ceramide is converted into sphingosine through a reversible reaction by ceramidase (CDase) and ceramide synthase (CerS) [18]. Sphingosine frequently converted to sphingosine-1-phosphate via the two sphingosine kinases (SPHK1/2) [19]. S1P is metabolised back to sphingosine via sphingosine phosphate phosphatases (SPPase) or by an irreversible reaction by S1P lyase [19]. S1P is then exported out of the cell via spinster 2 (SPNS2). Created in Biorender.com.

**Figure 2 ijms-24-06935-f002:**
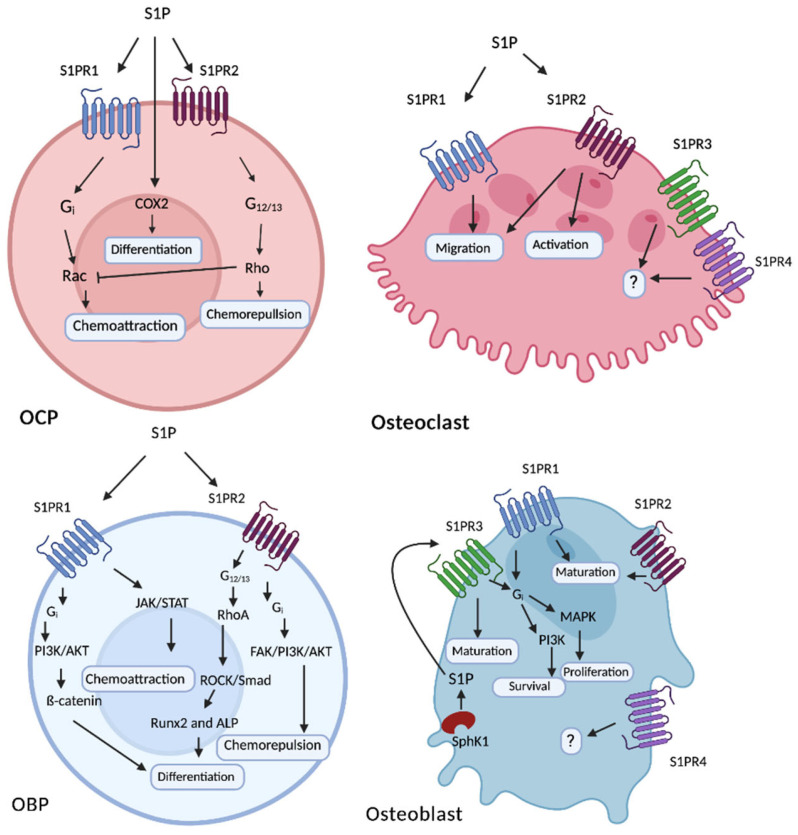
**S1PR function in osteogenic cells.** Schematic representation of the downstream signalling from S1PRs in osteoclast precursors (OCPs), osteoclasts (OCs), osteoblast precursors (OBPs), and osteoblasts (OBs). S1PRs are G-protein coupled receptors that activate various pathways and are present at different stages of OB and OC differentiation [17]. Within OCPs, S1PR1 and S1PR2 alter OC maturation via activation of Rho and Rac signalling pathways [24]. S1P can increase OCPs differentiation via Cox2, although the mechanism is not fully understood [24]. Although S1PR1-4 are expressed in OCs, the role of S1PR3 and S1PR4 are not currently understood [30]. S1PR1/2 can alter OC migration but the mechanisms underpinning this remain unclear. S1PR2 appears to be essential for OC activation. Within OBs, S1PR1 mediates OB differentiation via PI3K/AKT signalling and regulates migration via JAK/STAT signalling [31], resulting in chemoattraction in response to S1P [31], whereas S1PR2 increases OB differentiation via RhoA/ROCK/Smad signalling [30] and can also act through FAK/PI3K/AKT signalling to promote chemorepulsion. In OBs, S1PR1 increases OB proliferation [32] and survival via MAPK and PI3K signalling. S1PR1-3 are involved in mediating OB maturation through as of yet unknown signalling pathways. Activation of SphK1 in OBs promotes osteoblast maturation via autocrine S1P signalling through S1PR3 [23]. The function of S1PR4 in OBs is currently unknown. Created in Biorender.com.

**Figure 3 ijms-24-06935-f003:**
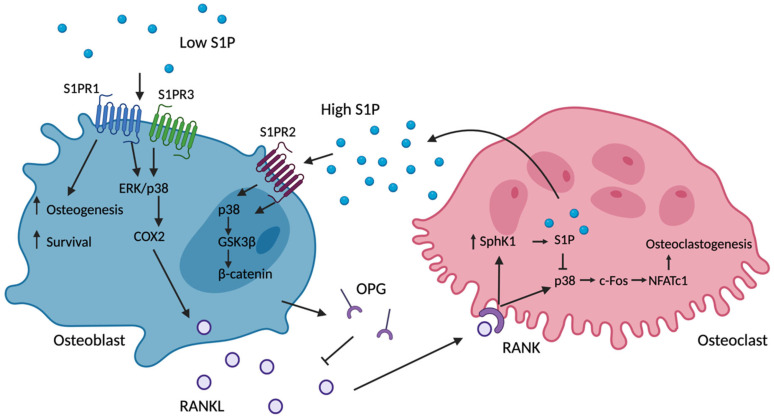
**Osteoblast–Osteoclast Communication via S1P.** S1P can act directly on osteoblasts via S1PR1/3 receptors to enhance osteoblast maturation and mineralisation [47]. S1PR1/3 also activates the ERK/p38 signalling pathway to enhance COX2 production [24,27]. COX2 then elicits the release of RANKL from osteoblasts to enhance osteoclastogenesis via p38-cfos-NFATc1 signalling [24]. RANK activation of osteoclasts by RANKL also upregulates SPHK1 expression and production, resulting in increased intracellular S1P, which inhibits p38. Inhibition of p38 causes a reduction in osteoclastogenesis, acting as a negative regulator of osteoclast maturation [24]. S1P released from osteoclasts increases extracellular S1P, which may act as a break on osteoclastogenesis through the S1PR2 receptor on osteoblasts, as S1PR2 acts to enhance the secretion of the RANKL decoy receptor osteoprotegerin (OPG) [40]. OPG acts to reduce further RANKL-mediated osteoclastogenesis through binding to RANKL. Created in Biorender.com.

**Figure 4 ijms-24-06935-f004:**
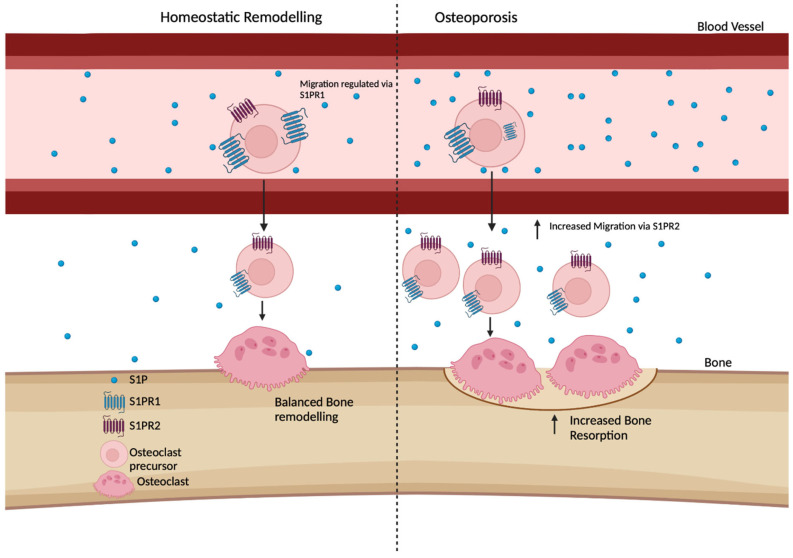
**Osteoclast precursor migration is increased in osteoporosis due to increased plasma S1P.** During typical homeostatic remodelling, it has been suggested that the S1P gradient allows for regulated migration of osteoclast precursors into the bone tissue via S1PR2 [35,36]. This is regulated via S1PR1, which opposes S1PR2, keeping some cells within circulation. Osteoporotic patients express increased plasma S1P [64,65,66]. This is thought to result in internalisation of the regulator receptor, S1PR1, causing increased migration of cells due to reduced opposition to S1PR2. Due to increased S1PR2 activity on osteoclast precursors, there are more migrating precursors that are thought to result in an increase in osteoclast numbers within the tissue, resulting in increased bone resorption. Created in Biorender.com.

**Table 1 ijms-24-06935-t001:** Summary table outlining the effects of S1P on the different stages of bone remodelling and their potential as therapeutic targets in catabolic bone disorders.

Cell Type, Function	Effect of S1P Signalling	Therapeutic Application in Catabolic Bone Disorders	Ref
Osteoblast migration	S1PR1 and S1PR3 mediate chemoattraction to S1P—supports migration of precursors to the bone surface. S1PR2 has dual actions: low concentrations promote migration towards S1P. High concentrations cause chemo-repulsion.	Direct targeting of receptors is contraindicated due to expected off-target effects on osteoclasts. Targeting of downstream signalling could prove effective.	[43,44]
Osteoblast proliferation	Exogenous application of S1P increases proliferation. Inhibition of S1PR2 enhances proliferation but not differentiation.	Inhibition of S1PR2 enhances total osteoblast number but not their maturation. Unlikely to be effective therapeutically.	[41,52]
Osteoblast differentiation	S1P increases osteoblast differentiation. S1PR2 is required for osteoblast differentiation and maturation. Inhibition reduces differentiation. S1P signalling via S1PR3 is required for bone formation and maintenance.	S1PR3 is an interesting potential therapeutic target as (unlike, e.g., S1PR2) it has no alternative effects on osteoclasts. S1PR3 agonists likely to increase osteoblast differentiation and increase bone formation.	[29,41,47,50,51,53]
Osteoclast migration	S1PR1 is required for chemoattraction to S1P, maintaining osteoclast precursors within the circulation. S1PR2 is required for chemorepulsion towards S1P, promoting osteoclast precursors migration to tissue.	Inhibition of S1PR2 via antagonists (e.g., JTE013), may lower the number of osteoclasts present within bone tissue, reducing resorption.	[36,37,39,40]
Monocyte fusion	S1PR2 is required for monocyte fusion into osteoclasts, via regulation of podosome-adhesive proteins.	Inhibition of S1PR2 via antagonists (e.g., JTE013) prevents osteoclast formation, reducing resorption.	[40]
Osteoclast differentiation	SPHK1 is required for negative regulation of RANKL-mediated differentiation, through suppression of p38 signalling. The impact of exogenous S1P on osteoclast differentiation is unknown (incomplete data available).	Therapeutics that upregulate SPHK1 function or suppress p38 signalling may reduce mature osteoclasts.	[24,48]
RANKL production	Activation of S1PR1/3 on osteoblasts induces RANKL release, via activation of ERK/p38.	Reduction of RANKL-mediated osteoclast differentiation could be achieved through inhibition of osteoblast S1PR1/S1PR3 receptors or targeting the downstream signalling pathways.	[23]
OPG Production	Activation of S1PR2 on osteoblasts induces OPG release via GSK3β and β-catenin.	S1PR2 agonists or targeting of β-catenin-mediated secretion of OPG may reduce osteoclast numbers to prevent bone resorption.	[23,35,54]

## Data Availability

No new data have been created.

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
