# Peer review of "The Ying and Yang of Sphingosine-1-Phosphate Signalling within the Bone"

_ijms, 2023, doi:10.3390/ijms24086935_

Round 1

Reviewer 1 Report

Dear editor and authors,

This is a well-prepared review that adequately discusses the accumulating evidence for the different, and in certain circumstances opposing, roles of S1P in bone homeostasis and disease, including osteoporosis, Pagets disease, and inflammatory bone loss. From this review, the authors conclude that S1P may be an effective biomarker of bone disease and also an attractive therapeutic target for disease. However, it is noted that there are still a few uncertainties that need to be further discussed. Here are some minor revisions considerations as follows:

Comments 1:

Page 1-Line 24

The introduction is wrongly written in the abstract. 

Comments 2:

Page 5-Line 152:

The figure on this page should be figure 2 rather than figure 1.

Comments 3:

Page 4-Line 3: “0.8 or thicker than 3mm are more prone to perforation”, 0.8 should be followed unit mm, 3m should be added space between 3 and mm.

Comments 4:

Page-12-line480:

The subtitle The good, the bad and the ugly of S1P in bone diseases confused me.

Could the authors explain the meaning of this title, and the differences between bad and ugly of S1P in bone diseases?

Comments 5:

Page-15-line 623:

Periodontitis is insufficiently discussed in this paragraph, there are many risk factors such as smoking, aging, and poor oral hygiene which may lead to periodontitis. Could the author check whether there is a relationship between risk factors and  S1P signaling?

Page-15-line 631:

“Of note, periodontitis is a risk factor for the development of RA, amongst other conditions, and as such aberrant S1P signalling within these individuals could contribute to systemic bone damage”

There is no reference to support this conclusion.

Also, the conclusion  It is important that further research is carried out to clarify the role and importance of S1P in RA and periodontitis to understand the effect and best method of targeting S1P in the treatment of inflammatory bone loss. is abrupt and confusing to the authors.

Comments 6:

Page-16-line 638:

The author suggests any future studies be carried out to fill these unknown assumptions, however, could the author discuss the difficulties of deploying and implementing these experiments around S1P?

Comments 7:

Page 16-Line 639:

The conclusion part is slightly verbose, if the author could refine this part, the readers may sooner get the highlights of this review.

Author Response

Pleasee see the attachment

Reviewer 3 Report

This review, which includes the adaptation of the Sphingosine 1 Phosphate signal in the bone within the framework of Ying and Yang, has been defined and compiled according to previous and current theoretical background and scientific research on the subject. 28 of 92 references were obtained from studies conducted between 2017-2022. In this case, the study is an informative and current review. English language and style in some parts of the review require fine/minor spelling, so authors should check thoroughly to make these minor corrections. Then this study will be suitable for publication.

Round 2

Reviewer 2 Report

I am Ok with revision.